# Brownian Motion in Optical Tweezers, a Comparison between MD Simulations and Experimental Data in the Ballistic Regime

**DOI:** 10.3390/polym15030787

**Published:** 2023-02-03

**Authors:** Krzysztof Zembrzycki, Sylwia Pawłowska, Filippo Pierini, Tomasz Aleksander Kowalewski

**Affiliations:** 1Department of Biosystem and Soft Matter, Institute of Fundamental Technological Research, Polish Academy of Sciences, ul. Pawinskiego 5B, 02-106 Warsaw, Poland; 2Faculty of Electronics, Telecommunications and Informatics, Gdańsk University of Technology, ul. G. Narutowicza 11/12, 80-233 Gdańsk, Poland

**Keywords:** Brownian motion, molecular dynamics, optical tweezers, ballistic regime, water model comparison

## Abstract

The four most popular water models in molecular dynamics were studied in large-scale simulations of Brownian motion of colloidal particles in optical tweezers and then compared with experimental measurements in the same time scale. We present the most direct comparison of colloidal polystyrene particle diffusion in molecular dynamics simulations and experimental data on the same time scales in the ballistic regime. The four most popular water models, all of which take into account electrostatic interactions, are tested and compared based on yielded results and resources required. Three different conditions were simulated: a freely moving particle and one in a potential force field with two different strengths based on 1 pN/nm and 10 pN/nm. In all cases, the diameter of the colloidal particle was 50 nm. The acquired data were compared with experimental measurements performed using optical tweezers with position capture rates as high as 125 MHz. The experiments were performed in pure water on polystyrene particles with a 1 μm diameter in special microchannel cells.

## 1. Introduction

The knowledge of interactions of colloidal particles suspended in water with their surrounding medium is crucial to understanding their behavior in crowded environments, such as cells and other biological materials [1,2], and diffusion is one of the most important processes taking place everywhere where there are fluids or gasses involved [3,4,5]. Often, molecular dynamics is used for simulating such conditions. Unfortunately, water molecule behavior is complex, and there are many theoretical models that try to reproduce it [6,7,8,9]. Some are not suitable for simulating the diffusion of particles in water, but many are, and choosing the correct one can be an arduous task. Another critical factor is the time scale of the simulation. Since molecular dynamics is computationally intensive, only relatively small systems or short time scales are actively computed. Unfortunately, the process of diffusion during a very small-time frame is very different from what is generally observed on macro scales; the movement of the particle becomes more ballistic than random, which means that below a characteristic relaxation time τ, the diffusing particle retains some of its momentum. This is called the ballistic regime, and the characteristic time can be calculated using the following Equations (1) and (2) [10]:(1)$$\tau =\frac{m+\frac{m_{f}}{2}}{\gamma }$$
(2)γ=6πηR
where:

*m*—particle mass

*mf*—mass of the fluid displaced by the particle when it moves

*γ*—friction coefficient

*η*—dynamic viscosity of the fluid

*R*—particle radius

A 1 μm diameter polystyrene colloidal particle has a relaxation time τ of about 93 ns, and during this time, it travels around 1 Å. If we reduce the diameter to 50 nm, the relaxation time goes down to 0.21 ns. Knowing the behavior of colloids in such short time scales has become very important in recent studies [11,12]. Another critical factor is that a biological system, similar to most others, is full of charged particles or particles that are polarized, such as water itself; therefore, electrostatic interactions will play a major role in the system.

Four water molecules were selected for comparison, all of which include electrostatic interactions of charged atoms: SPC (simple point charge) [13,14], SPCE (extended simple point charge) [15,16], TIP3P (transferable intermolecular potential 3 point) [17,18], and TIP4P (transferable intermolecular potential 4 point) [19,20]. Those were chosen because they are designed to be used in conditions close to a normal temperature and pressure or those common in biological systems. All of them can be effectively described as rigid pairs with potential composed of Lenanrd–Jones (LJ) and Coulombic terms, with all but TIP4P having three sites of interaction and the latter having a fourth mass-less point charge. On the other hand, they differ in how the water particle is described, in terms of parameters, and can vary quite substantially in computational complexity, which is reflected by the time required for a given simulation. All of them are used extensively to model the behavior of biological systems [21,22]. 

Since their discovery in 1970, optical tweezers have emerged as a tool to trap and manipulate nano and micrometer-sized material using a highly focused laser beam [23,24,25]. It is an extremely sensitive and precise instrument capable of manipulating objects and detecting their position with sub-nanometer precision and measuring forces with femtonewton (10–15 N) accuracy [26,27]. Now, they are finding applications in many fields of science, such as biology and chemistry, where they are used extensively in studying the unfolding of proteins [28,29,30]. Thanks to the advancements in the field electronics, it is now possible to further increase the resolution of optical tweezers in the time domain as well and observe the phenomena that happen on even smaller time scales in the range of ns [31,32]. Therefore, this technique is the best choice to measure the motion of colloidal particles with both time and spatial scales that match those achieved in numerical simulations.

An experimental study of the diffusion of polystyrene particles trapped in optical tweezers was conducted. This technique allows very precise measurements with high temporal and spatial resolution, even in the ballistic regime [33,34,35]. This allows a comparison of the results of numerical simulations with experimental results more directly and therefore can verify which water models are best suited for this range of applications.

## 2. Materials and Methods

In this study, use was made of both numerical simulations and experimental measurements of a spherical polystyrene bead moving in water. For time steps Δt>>τ, this is called Brownian motion. These fluctuations are stochastic in nature, and the general equation for the position as a function time is shown in the following differential Equation (3) [36]:(3)$$\dot{x}\left(t\right)=\sqrt{2k_{B}T\gamma }W\left(t\right)$$
where:

*x*—position

*t*—time

*k_B_*—Boltzmann constant

*T*—temperature

*W*(*t*)—white noise

The white noise term is not a standard function, but rather a process characterized by the following properties: its mean *<W(t)>* is equal to 0 for all values of tMean Squared Displacement (MSD) *<W^2^(t)>* is equal to 1 for all values of *t**W(t_i_)* and *W(t_j_)* are independent for all i ≠ j

If an external harmonic potential is applied to the particle, as generated by an optical trap, with the force if defined by Equation (4),
(4)$$F\left(x\right)=-kx$$
where:

*F*—force

*k*—force constant

then Equation (3) for the particle position takes the following form (5):(5)$$\dot{x}\left(t\right)=\frac{-k}{\gamma }x\left(t\right)+\sqrt{2k_{B}T\gamma }W\left(t\right)$$

The following Equation (6) ties the MSD to the diffusion coefficient, where *N* is a constant that depends on the degrees of freedom and takes vales of 2, 4, and 6 for 1, 2, and 3 degrees of freedom respectively.
(6)$$D=\frac{MSD}{Ndt}$$

The Stokes–Einstein Equation (7) can be used to calculate the theoretical diffusion of a colloidal particle in a stationary fluid without external force fields.
(7)$$D=\frac{k_{B}T}{\gamma }$$

As stated earlier, the above equations are true tor time steps Δt >> τ, where the MSD is proportional to t. In the ballistic regime, the MSD is proportional to t^2^ [34]. It is important to note however that in the Euclidean space, the positions of the particle in each exes are statistically independent; therefore, it is possible to compare diffusion coefficients obtained by from sources with different dimensions or degrees of freedom. For the purpose of this article, that is to compare MD water models, we will be calculating the diffusion coefficient, keeping in mind that in short time scales and in the presence of the trapping potential, it is not equal to the Stokes–Einstein relation. This is performed because the diffusion coefficient, as described by Equation (6), normalizes the MSD to the time step, making it easier to compare results obtained for different measurements.

### 2.1. Simulations

The LAMMPS (Large-scale Atomic/Molecular Massively Parallel Simulator, version 22 August 2018) [37] was used to perform simulations using all of the selected models [38,39]. It is one of the most commonly used simulators in many fields of science. The simulation consisted of a 2-dimensional rectangular box, 350 × 350 nm in size, filled with water, with a colloidal bead in the center. The box boundary in the simulation plane was periodic, meaning that particles interact across the boundary and can exit one end of the box and re-enter the other. The third unused dimension was fixed to 0. All molecules were able to freely rotate within the plane of simulation. Initially, the water molecules were placed on a rectangular grid with 3.1 Å spacing, which is equal to their normal mean distance [40]. The potential describing the molecule interactions was a standard Lennard–Jones given by the following Equation (8):(8)$$E\left(r\right)=4\epsilon \left[{\left(\frac{\sigma }{r}\right)}^{12}-{\left(\frac{\sigma }{r}\right)}^{6}\right]$$
where:

*E*—potential energy

*r*—distance between atoms

*ε*—depth of the potential well

*σ*—distance at which the intermolecular potential is zero

To calculate the LJ interaction parameters between different types of atoms, the Lorentz–Berthelot rules where applied, given by the following Equations (9) and (10)
(9)$$q_{ij}=\frac{q_{i}+q_{j}}{2}$$
(10)$$\epsilon_{ij}=\sqrt{\epsilon_{i}\epsilon_{j}}$$

The Coulombic pairwise interactions were given by the following Equation (11):(11)$$E_{ij}\left(r\right)=\frac{Cq_{i}q_{j}}{\epsilon r}$$
where

*E_ij_*—potential energy the between *i*-type and *j*-type atom

*r*—distance between atoms

*ε*—dielectric constant

*q*—charge of atom

*C*—energy conversion constant

The exact parameters of the molecule depend on the model and are shown in Table 1.

In all cases, the cutoff distance, above which the interaction forces were not calculated, was set to 10 Å for LJ and 15 Å for Coulombic interactions. For calculating the electrostatic interactions, the Particle–Particle Particle–Mesh (PPPM) method was set and the default velocity–verlet method was employed for LJ interactions. A round disc was placed in the middle of the simulation box, making the colloid particle. Since polystyrene is often used to make colloidal particles, the disc was made to simulate this material. Its composition was carbon and hydrogen in the exact same proportions as in a polystyrene chain. The atoms were placed in ring patterns, filling the entire disc, and a spacing corresponding with polystyrene, 1.4 Å between carbon atoms and 1.1 Å between hydrogen, and 1.8 Å spacing between carbon rings so that the overall density of the colloid would be the same as water in the simulations, which is 1 g/cm^3^. A real polystyrene bead used in the experiments has a slightly higher density of 1.005 g/cm^3^, but this difference is insignificant in the time scales discussed in this paper. The overall diameter of the disc was 50 nm, and it was configured to be one rigid body and interacted with water with the same LJ and Coulombic potential with parameters listed in Table 1. The size of the void space between colloid and water atoms was set so that the overall mass density of the system was also equal to 1 g/cm^3^, and the overall atom count was almost 4 million, with more than 1.25 million water molecules. To simulate an optical trap, a harmonic-type force was applied to the center carbon atom of the particle. At the beginning of each simulation, the water molecules were assigned velocities corresponding to a temperature of 298.0 K, generated with a random number generator. During the whole run, an NVE thermostat was applied to the whole system

### 2.2. Experiment

The optical trap [41,42] used in the experiments was developed earlier [43], but for the purpose of these measurements, the electronic front end had to be modified to increase its bandwidth. The detector was a quadrant avalanche photodiode (QA4000, First Sensor AG, Berlin, Germany), and the signals were digitized using a PC acquisition card (OctaveCSE8349, GaGe Vitrek LLC, Lockport, IL, USA), with a 125 MHz maximum sample rate, which equals a 8 ns time between individual samples, which is in the same order of magnitude as the overall simulated time. Experiments were conducted in microchannels made from PDMS, using a photolithography-based technique, with a geometry that consisted of a main channel and a series of round wells in which the measurements made were connected to the main channel by a single smaller one. This solved the problem of particle movement being influenced by stray fluid flow. All measurements were performed above 5 μm from the bottom channel wall in order to ensure that the effect of slip was negligible. Before the experiment, each PDMS microchannel was placed in a vacuum for 15 min and immediately filled with water. That caused any air bubbles in the wells to be removed. Polystyrene particles used in the measurements were 1 μm in diameter (R1000 Thermo Fisher Scientific, Waltham, MA, USA) and were dispersed in distilled water with a 1:10,000 concentration. The optical tweezers were calibrated to the same particle type. A fresh solution was made before every experiment to prevent aggregation and sedimentation.

## 3. Results and Discussion

The simulations were split into three groups based on the applied harmonic potential simulating the optical tweezers. First, it was set to zero, corresponding to a freely moving particle; next, was a force constant of 1 pN/nm, and lastly, 10 pN/nm, but only for the two most basic models, SPC and SPCE. Figure 1 shows a scaled-down version of the simulation box.

Due to computational complexity and real-time constraints, the TIP4P model was simulated only with a freely moving particle, without simulations that included a harmonic potential, similar to that in the case of other models discussed here. Each group consisted of four simulation runs, identical in parameters, the only difference being the random number generator seed that generated the initial water velocities. The value of the time step was different for each model and condition and was fine-tuned to the highest value achievable for the simulation to be stable. High values resulted in lost atoms, that is, atoms that escaped the simulation box or the temperature of the system exceeding the set value. The velocity of the colloid particle was set to 0 at the start of the run, and its position was printed every 1000 steps, and the time step value varied depending on the water model used. In all cases, at least 10 M steps were computed, which corresponds to about 30 ns of real time in most cases, depending on the actual time step setting. The equilibration phase was very short, and each system reached a steady state after about 10 k time steps. Each simulation was run on an identical computer equipped with two Intel Xeon X5650 processors with six (physical) cores running at 2.67 GHz and 24 GB of RAM. Table 2 contains the summarized model performance and resource usage. 

When an external force field was applied, in some cases, the time step value had to be modified for the simulation to be stable. However, in all cases, it did not impact the time step calculation time or memory usage. Figure 2 shows a typical particle movement, one from each model. Due to variations in the time step value, the overall simulated real time varied depending on the model and whether external force field was applied. 

From the computational perspective, both SPC and SPCE models behaved very similarly, having the same memory usage, and the same time step value and application of additional force field did not impact their performance. The only major difference is that the SPC model showed results slightly faster. On the other hand, the TIP3P and TIP4P behaved very differently depending on whether the additional force field was applied. In the latter case, its performance was not much worse than that of the SPC(E) models, but when an external force field was applied, the maximum time step value decreased dramatically by an order of magnitude. This significantly increased the need for resources required to compute a given amount of real time and was the primary reason why in this experiment, the TIP4P model was used only to simulate a freely moving colloidal particle.

Figure 3 shows calculated diffusion coefficients using Equation (12), which is a special case of Equation (6):(12)$$D=\frac{\langle ds^{2}\rangle }{4dt}$$
where:

*ds*—particle displacement

*dt*—time of the displacement, 1000 time steps in our case

Most considered models give an order of magnitude of higher diffusion than the theoretical value of 1.45 × 10^−11^ m^2^/s; only TIP4P is very close but is the most complicated model. As expected, with force applied, the diffusion drops, and the particle is confined close to the center of the simulated optical tweezers, with all models giving similar results. In the performed simulations, although very similar to those of SPCE, the SPC model had a faster computation time and yielded similar results with exactly the same memory usage. Employing the TIP3P model was more time-consuming and slightly more memory-consuming while giving similar results. It also had a lower time step value; thus, it required more computation time to simulate the same amount of real time. The model that produced the most accurate result was TIP4P, but on the other hand, it was significantly more resource-consuming, taking about 3.3 times the time to compute a time step and requiring twice the memory. 

**Figure 3 polymers-15-00787-f003:**
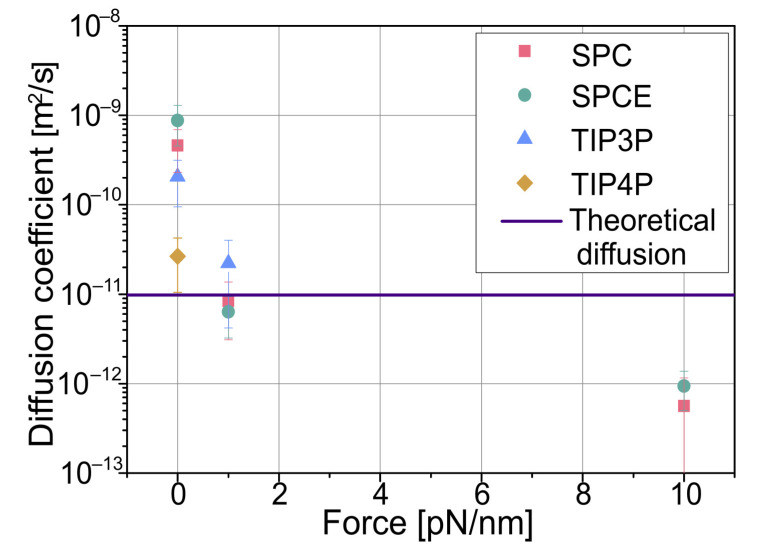
Calculated average diffusion coefficients obtained from simulation data. For the given colloidal particle, the theoretical diffusion coefficient without external force, calculated using the Stokes–Einstein equation, is 9.81 × 10^−12^ m^2^/s.

A series of experimental measurements was made on 1 μm polystyrene beads suspended in water trapped in optical tweezers with a stiffness of 1 pN/nm, the same as in the simulations. Multiple different particles were measured in order to eliminate errors caused by particle size differences. Every measurement was made in a special experimental chamber with only one inlet, the particle being dragged in by the optical tweezers. This technique ensured that there was no fluid movement caused by residue pressure, fluctuations in temperature, and mechanical vibrations that could cause additional forces influencing the particle and thus cause measurement errors. Figure 4 shows a picture of one such chamber.

Each particle was measured twice, first with a low sample rate of 1k samples/s and the second time with a maximum acquisition rate of 125 M samples/s. Care was taken to ensure that only one particle was within the tweezers. While hard to see under the microscope, the diffraction pattern of the laser light back-scattered from the objects within the optical trap is very different that when more than one particle is trapped. Results, calculated using the same formula as with simulations (6), are shown in Figure 5. The theoretical diffusion was calculated using the Stokes–Einstein Equation (13):(13)$$D=\frac{k_{B}T}{6\pi \eta R}$$
where:

*T*—absolute temperature, 298 °C in our case

*η*—dynamic viscosity, 0.89 mPa s^−1^ in our case

*R*—radius of the particle, 0.5 μm in our case

**Figure 5 polymers-15-00787-f005:**
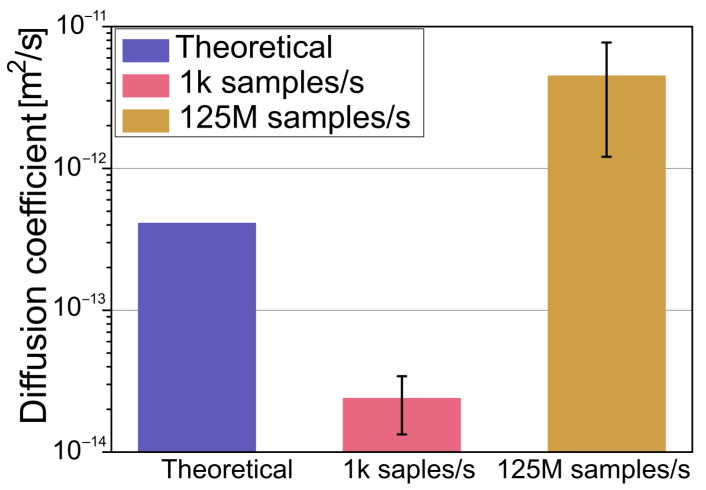
Measured diffusion of 1 μm polystyrene particle in an optical tweezers system with a 1 pN/nm trap stiffness, along with theoretical calculations for a freely moving particle.

As observed, the measured diffusion with a slow sample rate is smaller than that of a theoretical calculation for a freely moving particle. For comparison, the diffusion measured with a fast sample rate is higher. This is because the time scale is in the ballistic regime, where the mean square displacement has a nonlinear relationship with time. This increase in the measured diffusion coefficient in the ballistic regime is in agreement with the numerical simulations discussed in the previous paragraph. Although the particle used in the experiments is much bigger than that used in the simulations, as can be seen in Equation (7), the diffusion coefficient is independent of mass and is inversely proportional to the radius. Therefore, it is possible to qualitatively compare the obtained results.

The obtained results of simulations can be compared to the work of Pekka and Lennart [44] where the authors compared the MD simulations of water molecules simulated with SPC, SPCE, and TIP3P models. Some of the simulation parameters are also very similar to those used in this work, such as the time step (2 fs), temperature (298 K), and cutoff distance (12 Å). Results obtained by Pekka and Lennart differ based on the model used, almost by an order of magnitude, and all give higher values than those obtained by experiments, as was the case in this study. Yet, there are also major differences between the conducted simulations, mainly the scale; there, the box contained around 1k atoms compared to almost 8M in this work. Moreover, here, the diffusion of a polystyrene particle, with an external force applied, was simulated compared to pure water molecules. 

In the work by Pancorbo, Rubio, and Domínguez-García [45] the authors studied Brownian motion in optical tweezers using 2D simulations, very similar to this work. The disk had a diameter of 1.9 μm, at a temperature of 295.9 K, at trap. Their results agree to within 1% of theoretical values, which is a very good result. Yet, there are many differences between their work and this. Firstly, the simulation was not molecular dynamics-based and the time is in the rage of milliseconds, so it is not possible to verify different water models and simulate Brownian dynamics in the ballistic regime. In addition, there is no mention of electrostatic interactions between water and the molecule simulated.

To the best of our knowledge, this is the first attempt at a large-scale MD simulation of Brownian motion of a charged colloidal particle in different water models. Yet, as is always the case, there are some limitations of the employed methodology. Firstly, everything was performed in 2D. It would be even better to perform 3D simulations, which would take into account the non-symmetric potential of the optical trap, which is different along the optical axis of the lens to the potential in the plane perpendicular to the optical axis. Another drawback is that the studied water models are one of the simplest ones. There are other more sophisticated ones, such as TIP5P [46], which may yield more accurate results. One of the most evident differences between the experiment and the simulation is the size difference between the colloids. In the MD, it was 50 nm, whereas in the experiment, it was 1 μm. While this issue is addressed in this work, it would be good to perform simulations and experimental measurements with exactly the same sized colloids. While it is possible to detect nanometer-sized objects with optical tweezers, it has not been performed with high acquisition rates due to the a very small signal available form such a small object.

## 4. Conclusions

This work compares the behavior of different water models in molecular dynamics simulations and verifies those data against experimental measurements. Four models, SPC, SPCE, TPI3P, and TIP4P, were evaluated in regards to their resource usage and from a qualitative perspective, all yielded satisfying results, with the SPC being the most economical, but from a quantitative point of view, the TIP4P proved to be the most accurate. Both the SPC and SPCE models yielded very similar results and required the same amount of memory, with SPCE being 14% slower on the same machine. Moreover, the addition of an external force field did not impact their overall performance. As expected, the slowest model out of the tested four was the TIP4P, being about 70% slower than SPC and requiring 100% more memory. Further, the TIP3P and TIP4P models required a reduction of the time step value by an order of magnitude when an external force was added, dramatically increasing the amount of time required to simulate a given amount of real time. This was not the case with the SPC and SPCE models, meaning that the use of TIP3P and TIP4P models can cause significant drawbacks in many applications requiring long-term simulations. All tested models had the same behavior with the particle diffusion increased in the ballistic regime. 

Experimental measurements, made using optical tweezers with a 1 pN/nm trap stiffness, show an increased diffusion coefficient in the ballistic regime compared to that of the standard one measured with a relatively slow sample rate. This expected behavior corresponds well with numerical data, in which the diffusion in the ballistic regime is orders of magnitude higher in larger timescales. The data obtained in our study can prove crucial, as The Brownian motion of confined particles is often used in calibration procedures of optical traps. Therefore if the measurements were taken with a high enough acquisition speed, the calibration would be incorrect. In recent years a new technique of measuring fluid viscosity through observations of diffusion in the ballistic regime has been reported [11], and the results of this study should help in better predicting the results of simulations of such systems.

Possible future improvements on the subject of this work would be to change the number of dimensions in MD simulations to three, increase the size of the colloid to better match experimental capabilities, and finally employ a more sophisticated water model, such as the mentioned TIP5P. While all this requires significantly more computational power, due to advances in technology, the required computers will become more accessible.

## Figures and Tables

**Figure 1 polymers-15-00787-f001:**
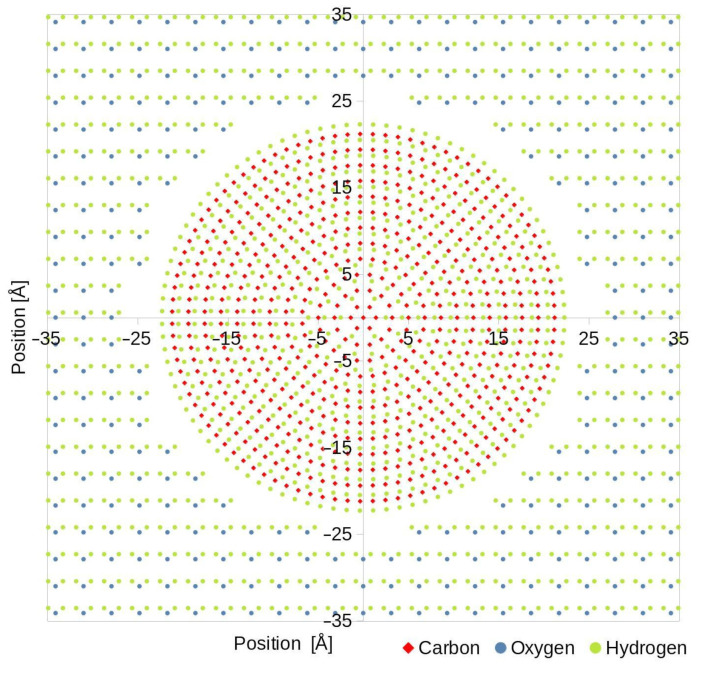
Scaled-down (computed simulation consisted of almost 4 million atoms) image of a simulation box filled with atoms. Water molecules were placed on a rectangular grid with 3.1 Å spacing, and in the center, was a solid particle composed of carbon and hydrogen atoms placed on a circular pattern.

**Figure 2 polymers-15-00787-f002:**
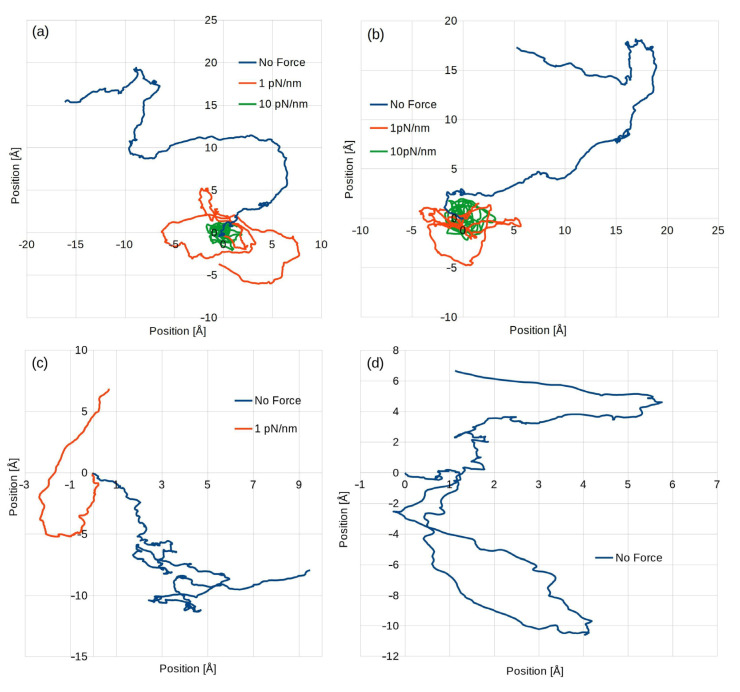
Sample traces of particles for models: SPC (**a**); SPCE (**b**); TIP3P, in this case, there was no simulation with a 10 pN/nm trap stiffness (**c**); and TIP4P, with only a freely moving particle (**d**).

**Figure 4 polymers-15-00787-f004:**
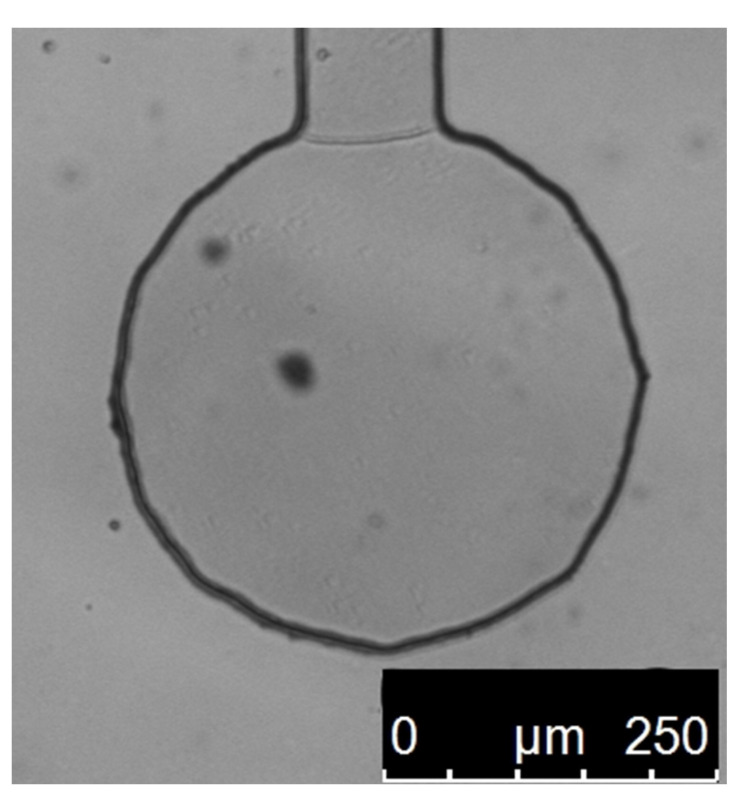
Picture of the experimental chamber, with one inlet, filled with pure water, in which the measurements on single particles were made. The inlet is at the top. It is part of a bigger microchannel chip made from PDMS, each chip containing around a dozen of such wells, each connected to one big channel (above the inlet, not shown here). Dark spots on the image are from dust on the optics and camera sensor.

**Table 1 polymers-15-00787-t001:** Parameters of atoms used in each simulation.

Water Model	SPC	SPCE	TIP3P	TIP4P
O-H distance	1 Å	1 Å	0.9572 Å	0.9572 Å
H-H angle	109.47 ^O^	109.47 ^O^	104.52 ^O^	104.52 ^O^
O charge	−0.82 e	−0.8476 e	−0.83 e	−1.04844 e
H charge	0.41 e	0.4238 e	0.415 e	0.52422 e
ϵ of O	0.1553 kcal mol^−1^	0.1553 kcal mol^−1^	0.102 kcal mol^−1^	0.16275 kcal mol^−1^
σ of O	3.166 Å	3.166 Å	3.188 Å	3.16435 Å
ϵ of C	0.644 kcal mol^−1^	0.644 kcal mol^−1^	0.644 kcal mol^−1^	0.644 kcal mol^−1^
σ of C	3.554 Å	3.554 Å	3.554 Å	3.554 Å
ϵ of H	0 kcal mol^−1^	0 kcal mol^−1^	0 kcal mol^−1^	0 kcal mol^−1^
σ of H	0 Å	0 Å	0 Å	0 Å

^O^—angle degrees, superscript. e—electron charge, normal, no superscript etc.

**Table 2 polymers-15-00787-t002:** Model performance and resource usage.

Model	Computation Time [Time Steps/s]	Memory Usage [GB]	Time Step Value with No Force [fs]	Time Step Value with Force Applied [fs]
SPC	0.179	9.8	3.25	3.25
SPCE	0.154	9.8	3.25	3.25
TIP3P	0.144	10	3	0.75
TIP4P	0.053	19.5	3	-

## Data Availability

The data that support the findings of this study are available from the corresponding author.

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
