# Peer review of "Brownian Motion in Optical Tweezers, a Comparison between MD Simulations and Experimental Data in the Ballistic Regime"

_polymers, 2023, doi:10.3390/polym15030787_

Round 1
Reviewer 1 Report (Previous Reviewer 1)
The paper can be published as it is
Author Response
Attached Response

Reviewer 2 Report (Previous Reviewer 2)
Authors present several models of diffusion which used for simulations and compared them with experimental data. The models are well known and standarly used by many other scientists. Their idea of comparison of models with experimental results is not new and original. On the other hand, the study provides comprehensive view on the problem. In my opinion, the study should be published after removing of type and space errors and settlement of other reviewes.
Abstract, line 12: measurement (not meas;urements)
Figs 3 and 5, y-axis: diffusion coefficient (not only diffusion)
Author Response
Attached Response

Reviewer 3 Report (Previous Reviewer 3)
The revised paper still lacks a well structured justification for both the numerical and theoretical models adopted to conduct the research and rationalize the results.
Author Response
Attached Response

Reviewer 4 Report (New Reviewer)
The authors need to include the more relevant mathematical formulations
The major contribution/novelty of the proposed work is missing in the paper.
The authors must need to include one separate section in the manuscript and in that section, they must need to discuss and presents a detail comparison with other reported works based on major achievements.
Grammatical errors must require to be rectified.
Author Response
Attached response

Round 2
Reviewer 3 Report (Previous Reviewer 3)
I appreciate the efforts of the authors to improve the manuscript by including a new paragraph describing previous works by Pekka et al. and Pancorbo et al.
Looking at the literature cited here, it seems that the simulations presented in this work represent the first 2D atomistic simulations of a disk in explicit water to investigate Brownian motion. If this is the case, the author should state it and comment about possible limitations of their model.
Some typos found in the file "polymers-2170814-peer-review-v2.pdf":
line 298 "between between"
line 300 "the the"
Author Response
Attached File

This manuscript is a resubmission of an earlier submission. The following is a list of the peer review reports and author responses from that submission.
Round 1
Reviewer 1 Report
I have been carefully reading your paper and have some major comments to address, plus some minor stuff. Let's start with the most important:
1) a big issue is represented by the fact that the size of the colloidal particle you have simulated is 50 nm, while the real particle used in the experiments is 1 micron across. In order to compare the results of the simulation with the experiments, you should make some reasonings about the mass/volume of the two, and consequently normalize the quantities (force, distance travelled, speed, etc.), reporting in the paper all of this.
2) a second point is about simulation statistics, which is nowadays fundamental in all the papers, and more frequently required by editors. It takes some extra work, but believe me, you will get a lot of citations, publishing a better job! So: running the code only once can give you an estimate of the distance travelled/speed, but we need here to compare numbers showing an average and a standard deviation, to do so you need to repeat at least 5 times for each code.
3) another fundamental issue is the fact that the experiment is run at one single force point (1 pN/nm), while simulations also are run at 10 pN/nm and zero. So my question is: can you compare the results of the experiment also at 10 pN/nm, and zero? If the same optical tweezer setup is used to measure scattering, in order to reduce the force field to zero, can it be defocussed for example?
Minor issues
Title: I suggest "experimental" instead of "experiential"
Abstract:
line 15: "model" instead of "contain"
line 22: TIP4P acronym not yet defined
Introduction:
36: actively
38: give exact definition of ballistic regime, including typical time scales, range of velocities and mean free path
44: define all the acronyms here used
54: it's
58: give time scales numbers
60: achieved
Table 1: why are the parameters slightly different in the four cases? explain
Materials and methods:
92/93: give density numbers, at your lab temperature
Experiment:
122: dispersed, not dissolved
Results and discussion:
138: "only simulated with a freely moving particle" this is not clear enough, rephrase
162: for each model please report all relevant numerical equations to compare them, and summarize the main differences, so that the reader can understand how they can be exploited!
Figure 2: yellow has a too low contrast, please change colour to increase readability
Figure 3: OK but as discussed above add real average and standard deviation to all of the points
Figure 4: I see two particles, not only one! can you comment on this?
Figure 5: what model is the theoretical one? specify. Also here, as discussed above, you should add error bar.
Author Response
Comment 1. A big issue is represented by the fact that the size of the colloidal particle you have simulated is 50 nm, while the real particle used in the experiments is 1 micron across. In order to compare the results of the simulation with the experiments, you should make some reasonings about the mass/volume of the two, and consequently normalize the quantities (force, distance travelled, speed, etc.), reporting in the paper all of this.
Response: Thank You for pointing this out. We have added an appropriate explanation in the discussion.
Comment 2. A second point is about simulation statistics, which is nowadays fundamental in all the papers, and more frequently required by editors. It takes some extra work, but believe me, you will get a lot of citations, publishing a better job! So: running the code only once can give you an estimate of the distance travelled/speed, but we need here to compare numbers showing an average and a standard deviation, to do so you need to repeat at least 5 times for each code.
Response: We acknowledge that it was not appropriately stated, along with the lack of error bars on figures, that we ran the simulations 4 times. It has been corrected. Unfortunately, due to limitations in computations capabilities, we could only run each simulation 4 times.
Comment 3. Another fundamental issue is the fact that the experiment is run at one single force point (1 pN/nm), while simulations also are run at 10 pN/nm and zero. So my question is: can you compare the results of the experiment also at 10 pN/nm, and zero? If the same optical tweezer setup is used to measure scattering, in order to reduce the force field to zero, can it be defocussed for example?
Response: Unfortunately, our optical tweezers setup is not capable of reaching 10 pN/nm, which was not known during the early stages of simulations. Also due to the two-laser design and the necessity to have good signal to noise ratio, reaching 0 pN/mn is also impossible with our experimental setup.
Minor issues.
Response: We believe that we have addressed all of the issues pointed above and would like to thank You very much for this very thorough review.

Reviewer 2 Report
The paper is focused onseveral water models in molecular dynamics for simulations of Brownian motion of colloidal particles. Simulations were carried out for "passive motion" - a freely moving particle - and one in a potential
force field with two different strengths. Results are compared with experimental data.
In my opinion, the study is interesting but (in spite of the comparison with experimental data) more theoretical. The goal of this study and a potential use of obtained results in practical application should be explained better.
Diffusion coefficients showed in Fig. 3 differ in magnitude (E-9 - E-12). No variance is showed. What is the reason of differences? I suppose that theoretical value 1.45E-11 m2/s is valid for self diffusion (without external force). Why the values for 0 pN/nm are much higher? What is error of their determination?
Authors should explain the above mentioned problems.
Author Response
General comment: The paper is focused on several water models in molecular dynamics for simulations of Brownian motion of colloidal particles. Simulations were carried out for "passive motion" - a freely moving particle - and one in a potential
force field with two different strengths. Results are compared with experimental data.
Response: We would like to thank the Reviewer very much for her/his positive feedback.
Comment 1. n my opinion, the study is interesting but (in spite of the comparison with experimental data) more theoretical. The goal of this study and a potential use of obtained results in practical application should be explained better.
Response: Thank You for pointing this out. This study is meant to be more of the theoretical nature but we have added more explanation about this problem and potential applications..
Comment 2. Diffusion coefficients showed in Fig. 3 differ in magnitude (E-9 - E-12). No variance is showed. What is the reason of differences? I suppose that theoretical value 1.45E-11 m2/s is valid for self diffusion (without external force). Why the values for 0 pN/nm are much higher? What is error of their determination?
Response: We have added appropriate text explaining how the results are calculated and added error figures.

Reviewer 3 Report
The authors performed atomistic molecular dynamics simulations of water in two dimensions in the presence of a two dimensional disc made of hydrogen and carbon atoms mimicking a colloidal particle. They considered four different types of water models and compared the corresponding extracted colloid's diffusion coefficients.
The authors further performed experiments of a much larger colloid in an optical tweezer and compared the experimental diffusion coefficient with the ones extracted from simulations.
The authors do not make any attempt to understand the physical origin of what they observed, thus failing the main objective of their work, which, if I understood correctly, is that of establishing which water model (or, I would suggest, which computational setup as a whole) is best suited to study this system.
Below are some comments that the author could use to improve their manuscript.
- Many simulation details are missing:
- How does water interact with the rigid disc?
- How were long range electrostatic and van der Waals interactions computed?
- How was the diffusion coefficient computed?
- Which thermostat was used?
- How was the density of water fixed? The system is two dimensional, but the authors state that the density is fixed to 1g per cm cube. This needs clarifications.
- The figure 1 shows a "scaled down" particle of diameter 50 Angstrom in a cell of side 70 angstrom. The text speaks of a 50 nm particle in a box of side 350 nm. This is confusing. The author could use any visualization software like VMD and show the actual simulation box, with an inset showing a close up of the atomic arrangement in the disc and surrounding water, if needed.
-I guess the water molecules are placed "flat" in the simulation box, with H atoms and O atom in the plane of the colloidal disc. This simply sounds wrong. The dipole moment of water should be allowed to rotate freely. How does this assumption affect the results? There are no comments on how the authors came up with this two dimensional model or discussions about its possible implications.
- Are there other numerical studies in the literature that demonstrate the ability of the 2D model used here to describe the experiments conducted here?
- Other details that would greatly help the reader are missing: total number of water molecules (and overall number of atoms), plots of equilibration phase, techniques used to compute observables and estimate statistical errors (which are not reported here), distance between periodic images of the model colloidal particles...
Author Response
The authors performed atomistic molecular dynamics simulations of water in two dimensions in the presence of a two dimensional disc made of hydrogen and carbon atoms mimicking a colloidal particle. They considered four different types of water models and compared the corresponding extracted colloid's diffusion coefficients.
The authors further performed experiments of a much larger colloid in an optical tweezer and compared the experimental diffusion coefficient with the ones extracted from simulations.
Response: We would like to thank the Reviewer very much for her/his positive feedback.
We are glad that the Reviewer appreciated this research and the structure of the article. We would also like to thank the Reviewers for her/his valuable comments and corrections aimed at improving the quality of the manuscript.
Comment 1. The authors do not make any attempt to understand the physical origin of what they observed, thus failing the main objective of their work, which, if I understood correctly, is that of establishing which water model (or, I would suggest, which computational setup as a whole) is best suited to study this system.
Response: Thank You for bringing this to our attention. We have made appropriate changes to the manuscript to address this issue.
Comment 2. Many simulation details are missing:
Response: We have added this information in the manuscript.
Comment 3. How does water interact with the rigid disc?
Response: We have added this information in the manuscript.
Comment 4. How were long range electrostatic and van der Waals interactions computed?
Response: We have added this information in the manuscript.
Comment 5. How was the diffusion coefficient computed?
Response: We have added this information in the manuscript.
Comment 6. Which thermostat was used?
Response: We have added this information in the manuscript.
Comment 7. How was the density of water fixed? The system is two dimensional, but the authors state that the density is fixed to 1g per cm cube. This needs clarifications.
Response: We acknowledge that it may be confusing. Since mean separation between particles of water is about 3.1Å we can imagine making a initial 3D rectangular grid (of given density) and then taking a slab out of it as our 2D simulation box. We have added this information in the manuscript.
Comment 8. The figure 1 shows a "scaled down" particle of diameter 50 Angstrom in a cell of side 70 angstrom. The text speaks of a 50 nm particle in a box of side 350 nm. This is confusing. The author could use any visualization software like VMD and show the actual simulation box, with an inset showing a close up of the atomic arrangement in the disc and surrounding water, if needed.
Response: The simulation consisted of almost 4M atoms, the image would be just a rectangular with a disc without any detail. We have added better explanation for this figure.
Comment 9. I guess the water molecules are placed "flat" in the simulation box, with H atoms and O atom in the plane of the colloidal disc. This simply sounds wrong. The dipole moment of water should be allowed to rotate freely. How does this assumption affect the results? There are no comments on how the authors came up with this two dimensional model or discussions about its possible implications.
Response: he water molecules are able to rotate, we have added this explanation to the manuscript. 2D simulations of such systems are common because of less computational requirements than full 3D simulations, even in recent years.
Comment 10. Are there other numerical studies in the literature that demonstrate the ability of the 2D model used here to describe the experiments conducted here?
Response: Yes, there are, over the years 2D diffusion simulations are very common, even recently for example:
Huang, Tao & Zeng, Chun-Hua & Chen, Yong. (2022). Collective diffusion in a two-dimensional liquid composed of Janus particles. Communications in Theoretical Physics. 74. 10.1088/1572-9494/ac8f3f.
Comment 11. Other details that would greatly help the reader are missing: total number of water molecules (and overall number of atoms), plots of equilibration phase, techniques used to compute observables and estimate statistical errors (which are not reported here), distance between periodic images of the model colloidal particles…
Response: We have added this information in the manuscript.
